# Disability and depression among stroke survivors attending rehabilitation facilities at three designated tertiary care hospitals in Bangladesh: A cross-sectional study

**Mohammad Jahirul Islam**[1‡], **Sohel Ahmed**[2,3‡]*, **Khandaker Md Kamrul Islam**[1], **Progya Laboni Tina**[4], **Ayon Deb Nath**[5], **Nipa Biswas**[6], **Md Shafiqul Islam**[7], **Bikash Juty Dey Shikder**[1], **Muhammed Abdullah Al Mamun**[1], **Nasima Yasmin**[8], **Shishir Ranjan Chakraborty**[1]

1 MAG Osmani Medical College Hospital, Sylhet, Bangladesh, 2 Ahmed Physiotherapy & Research Center, Kalabagan, Dhaka, Bangladesh, 3 Directorate of Student's Welfare, Bangladesh University of Engineering and Technology, Dhaka, Bangladesh, 4 Parkview Medical College, Sylhet, Bangladesh, 5 St Richard's Hospital, Spitalfield Lane, Chichester, United Kingdom, 6 National Institute of Preventive and Social Medicine, Dhaka, Bangladesh, 7 Chittagong Medical College Hospital, Chittagong, Bangladesh, 8 Gonoshasthay Somajvittik Physiotherapy College, Dhaka, Bangladesh

‡ MJI and SA are contributed equally to this work as share first authorship.
* ptsohel@gmail.com

**Data Availability Statement:** All relevant data are within the manuscript and its Supporting Information files.

## Abstract

### Background

Poststroke depression (PSD) is a highly prevalent and serious mental health condition affecting a significant proportion of stroke survivors worldwide. While its exact causes remain under investigation, managing PSD presents a significant challenge.

### Aim

This study aimed to evaluate the prevalence and predictors of depression among Bangladeshi stroke victims.

### Methods

A cross-sectional study was carried out with 725 stroke victims who were receiving medical care at three designated tertiary care hospitals in Sylhet from January to December 2023. Depression and disability were measured using the Patient Health Questionnaire-9 and the Modified Rankin Scale. Logistic regression analysis was employed to examine the predictors linked to depression.

### Results

According to the study, 80.8% of individuals had moderate to severe disability, and 58.1% of them experienced a moderate to severe level of depression. Individuals who had hemorrhagic stroke (AOR 1.31, 95% CI: 0.77–2.25), repeated episodes (AOR 3.41, 95% CI: 1.89–6.14), tobacco use (AOR 1.76, 95% CI: 1.16–2.67), or coexisting health conditions (AOR

**Funding:** The author(s) received no specific funding for this work.

**Competing interests:** The authors have declared that no competing interests exist.

1.68, 95% CI: 1.00–2.82) exhibited elevated levels of depression. Participants whose medical expenses covered by relatives or others were six times more likely to experience depressive symptoms (AOR 6.32, 95% CI: 1.61–24.76). Individuals who did not receive rehabilitation services had two times greater odds of being depressed (OR 1.85, 95% CI: 1.23–2.77, p = 0.003). Consequently, individuals with low functional status had eleven times greater levels of depression (AOR 11.03, 95% CI: 7.14–17.04).

## Conclusion

More than half of the participants in this present study reported moderate to extreme levels of depression which is a serious health issue among Bangladeshi stroke survivors. Understanding the predictors of depression linked to stroke could enhance the effectiveness of therapeutic interventions for this condition. In addition, multidisciplinary teams should work collaboratively to address this serious issue.

## Introduction

Stroke represents a significant public health crisis globally, ranking as the leading cause of disability and death worldwide [1]. The 2019 Global Burden of Disease (GBD) study highlights stroke as the second leading cause of death and the third leading cause of disability globally. There were approximately 12·2 million incident cases, 101 million prevalent cases, 143 million disability-adjusted life years (DALYs) lost due to stroke, and 6·55 million stroke-related deaths [2]. In Bangladesh, the stroke incidence rate is 11.39 per 1000 people [3]. The GBD study further indicated that stroke is the foremost cause of mortality and the second most common cause of disability within Bangladesh [4].

Stroke is an extremely unexpected event in a patient's life, leading to a wide range of psychological, social, and economic consequences, along with mental health challenges [5]. Among the various conditions leading to disability, poststroke depression (PSD) is the most prevalent and significant neuropsychiatric consequence following a stroke [6]. The incidence of depression following a stroke ranges from 25% to 79%, with more than half of these individuals going undiagnosed or untreated [7]. In the first year following a stroke, patients with PSD had greater functional impairment, longer hospital stays, worse rehabilitation outcomes, a lower quality of life, and a greater mortality rate [8–11]. Earlier studies reported that female sex, older age, severity of stroke, and living environment increased the risk of PSD [8, 10, 12, 13].

A Bangladeshi study revealed that 70% of poststroke patients developed depressive symptoms, with 32% experiencing more severe symptoms [14]. Three months after their stroke, 35.8% of stroke survivors in Sri Lanka reported having depression [15]. A systematic review reported that the pooled incidence of poststroke depressive symptoms in India was nearly 55% [16]. To develop prevention, early detection, and appropriate treatment methods that lead to better outcomes, it is imperative to gather information regarding the prevalence and predictors of depression among stroke victims. Unfortunately, sufficient data for Bangladesh are lacking, especially for people who have suffered stroke. This study focused on evaluating the prevalence of depression and exploring the relationship between disability and depressive symptoms among Bangladeshi stroke survivors which will help establish the urgency of intervention in this emerging field. Additionally, this study also identified predictors of depression among

Bangladeshi stroke survivors. The results are anticipated to play a crucial role in providing valuable insights for both clinical practices and health policy, ultimately enhancing the mental well-being of stroke survivors in Bangladesh.

## Methodology

### Study design and setting

A cross-sectional study was conducted using convenience sampling of stroke patients recruited from three tertiary-level hospitals in Sylhet, Bangladesh. The data were collected from January to December 2023.

### Study population and ethical considerations

This study enrolled stroke patients who received physiotherapy at three designated tertiary care hospitals during their regularly scheduled follow-up visits. The study adhered to the ethical standards set by the Bangladesh Medical Research Council and the 2013 Helsinki Declaration for human participant involvement. Ethical approval was granted by the Institute of Physiotherapy, Rehabilitation, and Research, the official institute of the Bangladesh Physiotherapy Association, under reference number BPA-IPRR/IRB/07/12/2022/109. Prior to their involvement in this investigation, we received written informed consent from the participants. For participants who were unable to read and write, their thumbprint was collected, and consent was sought from their legal guardians.

### Sample and sample size estimation

The sampling procedure for the study was performed using the following equation: $Z^2pq/d^2$. where $Z^2 = 1.96$, p = the expected proportion of depression reported in a prior Bangladeshi study in patients with stroke is 70% [14], d denotes the margin of error at 3.5%, and an attrition rate of 10% is included to minimize bias. Using this calculation, the study required a minimum sample size of 725.

### The eligibility criteria

**Inclusion criteria.** The study enrolled individuals aged eighteen and eighty years who had confirmed stroke attack, as evidenced by computed tomography (CT) or magnetic resonance imaging (MRI), occurring at least two weeks prior after the events began [17]. The participants expressed their willingness to participate in the study.

**Exclusion criteria.** The exclusion criteria for participants were prior musculoskeletal conditions, psychoactive substance addiction, existing mental illness before stroke occurrence, communication challenges, medical instability, various neurologic diseases and end-of-life situations [18].

### Data collection procedures

The data were collected using an interviewer-administered questionnaire that was meticulously sent to Bangla and subsequently translated to English. Our data collection process utilized the echo-friendly Google Forms platform as an alternative to traditional pen and paper methods. The inclusion of the mandatory option on each question serves to avoid incomplete submissions. The interviews were performed by three proficient data collectors, all of whom were physiotherapy graduates with more than three years of experience. Prior to data collection, a training session was conducted to demonstrate data collection process and maintain consistency among the data collectors. In order to address any issues that may arise during

data collection, each data collector carried out a pre-test session including a minimum of five data. These data were excluded from the final analysis. Clinical data were obtained from the patients' medical files, including details on the type of stroke, the side of paralysis, the length of the event, any concomitant disorders and whether they received rehabilitation services.

## Data collection tools

The questionnaire covers sociodemographic data, including age, sex, living location, education level, marital status, employment status, income, smoking and tobacco ingestion habits, living arrangements and caregiver support. The clinical section included details on the form of stroke, side of paralysis of the body, length of the event, concomitant disorders, and who was responsible for receiving medical treatment and rehabilitation services.

**The modified Rankin scale (mRS)** is commonly used to measure difficulties following a stroke, in accordance with the International Classification of Functioning, Disability, and Health or ICF model framework, which takes bodily function, activities, and social involvement into account. The tool has a 7-point system that extends from 0 (indicating problems and complete normalcy) to 5 (indicating serious handicap), with 6 representing dying. Importantly, the tool demonstrates strong test-retest reliability, typically ranging between k = 0.81 and 0.95 [19].

**The Patient Health Questionnaire-9 (PHQ-9)** was used to evaluate the levels of depression in the study participants. This rapid depression assessment tool is commonly used across several diagnostic categories and healthcare environments. Higher scores indicate more severe depression. The instrument demonstrated good specificity (0.75) and sensitivity (0.87) in screening and diagnosing severe depression, and a valid and reliable tool to measure post-stroke depression [20].

## Assessment of variables

**Independent variables.**   Age was initially treated as a continuous variable; however, it was later divided into two categories: ≤ sixty years and >sixty years. Additionally, the variables were organized as follows: gender (either male or female); living location (either urban or rural); marital status (either single/widow or married); educational level (either primary or secondary and higher); employment status (encompassing job holder, business, farmer, casual labor) or unemployed (including housewife, retirees, others); family income (monthly) (less than 15000 taka, between 15000–30000 taka or >30000 taka); smoking status and tobacco usage (yes or no); living arrangements (living with spouses and children or others); caregivers (spouses and children or others); stroke type (ischemic or hemorrhagic); length of the stroke event (predication into six months or shorter or more than six months); side of paralysis (right, left or both sides); stroke occurrence (first or multiple episodes); concomitant disorders (no or yes); and who was responsible for paying for medical treatment (spouse/children/others) and receiving rehabilitation (no or yes).

**Dependent variables.**   Patient Health Questionnaire 9 (PHQ-9): The final score is determined by adding up the scores of nine items, with a number between 0 and 27. Based on the sum of the number, the level of depression is defined in four stages as follows: minimal or no depression (0–4), mild depression (5–9), moderate depression (10–14), moderately severe depression (15–19), and severe depression (20–27). A score below 10 indicated the absence of depression, while a score of 10 or above indicated the presence of depression [21]. We categorized the presence or absence of depression as a binary outcome (yes or no) for the final analysis.

## Data analysis

The data analysis was performed with SPSS-25 software, encompassing both descriptive and inferential statistics. Continuous data were summarized through means and standard deviations (SDs), while categorical data were displayed via frequencies and percentages. Spearman's rank correlation coefficient was calculated to investigate the association between disability and depression among stroke patients. The chi-square test and Fisher's exact test were applied in univariate analysis to explore the relationships between sociodemographic factors, clinical features and depression. Binary logistic regression was used to identify potential risk factors for depression in stroke patients; independent variables with p values less than 0.05 in the univariate analysis were included, and adjusted odds ratios (AORs) with 95% confidence intervals (CIs) were used. The suitability of the model for predicting depression was investigated using the Hosmer–Lemeshow test and a classification table. The multicollinearity among the independent variables was evaluated using variance inflation factors (VIFs), with a cutoff point set at VIF≤5.0 [22]. The significance threshold was set by researchers at <0.05.

## Results

### Sociodemographic of the participants

As there was no missing data, this study includes a total of 725 people, including both males and females, participated in this study. The majority of participants were male (66.2%) and aged ≤60 years (61.2%). A total of 79.3% of the participants lived in rural areas, 82.8% were married, and 52.4% reported a monthly household income between 15000 and 30000 Bangladeshi Taka. Majority of the participants (75.4%) in the study group had a primary education or lower, and 50% of them were employed. A total of 81.0% of individuals had a stroke duration of ≤6 months, while 43.4% of participants received rehabilitation services. The results also reported that 84.0% of participants were diagnosed with ischemic stroke, while 19.2% experienced a repeat attack. The study patients had comorbid diseases, including hypertension (68.1%), diabetes mellitus (30.9%), and both diabetes mellitus and hypertension (26.6%). The information is provided in Table 1.

### Prevalence of disability and depression among stroke survivors

The results showed that 7% of participants had no symptoms at all, and 12.1% had no significant disability. Others had slight (18.8%), moderate (23.3%), moderately severe (27.7%), or severe (11.0%) disability. Among the study participants, 18.8% reported no depression, 23.2% had slight depression, 41.4% had moderate depression, 15.2% had severe depression, and 1.5% had extremely severe depression.

### Factors associated with depression among stroke survivors

Depression was significantly associated with sex (p = 0.012), age group (p = 0.022), residence (p<0.001), marital status (p = 0.004), education (p<0.001) and occupation (p = 0.049). This research also revealed a significant relationship between depression and monthly family income (p<0.001), tobacco consumption (p<0.001), duration of stroke (p = 0.020), side of stroke (p = 0.002), type of stroke (p = 0.029), onset of stroke (p<0.001) and rehabilitation service status (p = 0.001). The presence of concomitant disorders (p<0.001), medical expense provider (p<0.001), caregiver support (p<0.001), and cohabitation situation (p<0.001) were all significantly correlated with depression. No significant associations were found between smoking habits (p = 0.368) or the presence of diabetes (p = 0.197) in the present study.

**Table 1. Association between stroke victims' sociodemographic status, clinical profiles, level of disability, and poststroke depression.**

| Variable | Total sample (n, %) | No Depression (n, %) | Depressed (n, %) | P-value |
|---|---|---|---|---|
| Total sample | 725 (100) | 304 (41.9) | 421 (58.1) | |
| **Gender** | | | | 0.012* |
| Male | 480 (66.2) | 217 (45.2) | 263 (54.8) | |
| Female | 245 (33.8) | 87 (35.5) | 158 (64.5) | |
| **Age group in year** | | | | 0.022* |
| <60 | 444 (61.2) | 201 (45.3) | 243 (54.7) | |
| >60 | 281 (38.8) | 103 (36.7) | 178 (63.3) | |
| **Residence** | | | | <0.001** |
| Rural | 575 (79.3) | 221 (38.4) | 354 (61.6) | |
| Urban | 150 (20.7) | 83 (55.3) | 67 (44.7) | |
| **Marital status** | | | | 0.004* |
| Married | 600 (82.8) | 266 (44.3) | 334 (55.7) | |
| Unmarred/Divorced/Widow | 125 (17.2) | 38 (30.4) | 87(69.6) | |
| **Education** | | | | <0.001** |
| Primary or less | 547 (75.4) | 211 (38.6) | 336 (61.4) | |
| Secondary and above | 178 (24.6) | 93 (52.2) | 85 (47.8) | |
| **Occupation** | | | | 0.049* |
| Employed | 360 (49.7) | 162 (45.0) | 198 (55.0) | |
| Unemployed | 365 (50.3) | 142 (38.9) | 223 (61.1) | |
| **Monthly family income (in Bangladeshi Taka)** | | | | <0.001** |
| <15000 | 146 (20.1) | 41 (28.1) | 105 (71.9) | |
| 15000–30000 | 380 (52.4) | 164 (43.2) | 216 (56.8) | |
| >30000 | 199 (27.4) | 99 (49.7) | 100 (50.3) | |
| **Smoking habit** | | | | 0.368 |
| No | 396 (54.6) | 172 (43.4) | 224 (56.6) | |
| Yes | 329 (45.4) | 132 (40.1) | 197 (59.9) | |
| **Tobacco ingestion** | | | | <0.001** |
| No | 394 (54.3) | 196 (49.7) | 198 (50.3) | |
| Yes | 331 (45.7) | 108 (32.6) | 223 (67.4) | |
| **Duration of stroke** | | | | 0.020* |
| ≤6 months | 587 (81.0) | 234 (39.9) | 353 (60.1) | |
| >6 months | 138 (19.0) | 70 (50.7) | 68 (49.3) | |
| **Side of stroke** | | | | #0.002* |
| Right | 357 (49.2) | 149 (41.7) | 208 (58.3) | |
| Left | 354 (48.8) | 155 (43.8) | 199 (56.2) | |
| Both | 14 (1.9) | 0 (0) | 14 (1.9) | |
| **Stroke type** | | | | 0.029* |
| Ischemic | 609 (84.0) | 266 (43.7) | 343 (56.3) | |
| Hemorrhagic | 116 (16.0) | 38 (32.8) | 78 (67.2) | |
| **Onset of stroke** | | | | 0.001** |
| 1st onset | 586 (80.8) | 283 (48.3) | 303 (51.7) | |
| Recurrent | 139 (19.2) | 21 (15.1) | 118 (84.9) | |
| **Co-morbidity** | | | | 0.001** |
| No | 128 (17.7) | 72 (56.3) | 56 (43.8) | |
| Yes | 597 (82.3) | 232 (38.9) | 365(61.1) | |

(*Continued*)

**Table 1.** (Continued)

| Variable | Total sample (n, %) | No Depression (n, %) | Depressed (n, %) | P-value |
|---|---|---|---|---|
| **Diabetes** | | | | 0.197 |
| No | 501 (69.1) | 218 (43.5) | 283 (56.5) | |
| Yes | 224 (30.9) | 86 (38.4) | 138 (61.6) | |
| **Hypertension** | | | | 0.014* |
| No | 231 (31.9) | 112 (48.5) | 119 (51.5) | |
| Yes | 494 (68.1) | 192 (38.9) | 302 (61.1) | |
| **Diabetes with Hypertension** | | | | 0.177 |
| No | 532 (73.4) | 231 (43.4) | 301(56.6) | |
| Yes | 193 (26.6) | 73 (37.8) | 120 (62.2) | |
| **Pay for medical treatment by** | | | | 0.001** |
| Self/Spouse | 235 (32.4) | 124 (52.8) | 111(47.2) | |
| Children | 459 (61.7) | 176 (38.3) | 283 (61.7) | |
| Others (Siblings/Relatives) | 31 (4.3) | 4 (12.9) | 27 (87.1) | |
| **Caregiver** | | | | 0.001** |
| Spouse and Children | 520 (71.7) | 242 (46.5) | 278 (53.5) | |
| Others (Siblings/Relatives) | 205 (28.3) | 62 (30.2) | 143 (69.8) | |
| **living arrangements** | | | | 0.001** |
| Spouse and Children | 576 (79.4) | 261 (45.3) | 315 (54.7) | |
| Others (Siblings/Parents/Relatives) | 149 (20.6) | 43 (28.9) | 106 (71.1) | |
| **Undergoing rehabilitation service** | | | | 0.001* |
| No | 410 (56.6) | 138 (33.7) | 272 (66.3) | |
| Yes | 315 (43.4) | 166 (52.7) | 149 (47.7) | |
| **Level of disability** | | | | 0.001** |
| MRS <3 (Good functional status) | 275 (37.9) | 215 (78.2) | 60(21.8) | |
| MRS≥3 (Poor functional status) | 450 (62.1) | 89 (19.8) | 361 (80.2) | |

Statistically significant

*p< 0.05; Statistically significant

**p<0.001

# Fisher's Exact test

## Correlation between disability and depression

Spearman's rank correlation coefficient showed a statistically significant correlation between disability and depression (p<0.001) among the stroke survivors. According to the Portney and Watkins correlation coefficient values, this study revealed a strong positive correlation (r = 0.655) between disability and depression.

## Predictors of depression among stroke survivors

Female sex (OR 1.32, 95% CI: 0.71–2.44, p = 0.367), living in a rural area (OR 1.28, 95% CI: 0.73–2.25, p = 0.375) and being married (OR 1.28, 95% CI: 0.42–3.68, p = 0.661) were predictors of depression among stroke survivors. Individuals with a monthly income <15000 BDT had a significantly greater level of depression (OR 2.39, 95% CI: 1.21–4.71, p = 0.011). Tobacco use and comorbidity were found to be predictors of depression, with odds ratios of 1.76 and 1.68, respectively. Individuals who had experienced hemorrhagic stroke were approximately 1.31 times more likely to have depression (OR 1.31, 95% CI: 0.77–2.25, p = 0.314) than those who had experienced ischemic stroke. Participants who experienced recurrent stroke exhibited

a significantly greater level of depression (OR 3.41, 95% CI: 1.89–6.14, p <0.001) than did those who experienced their first stroke. Individuals whose medical expenses were covered by their children had a 32% reduced likelihood of experiencing depression (OR 0.68, 95% CI: -4.0–1.13 p = 0.140). Conversely, individuals whose medical expenses were covered by relatives or others had a sixfold increased likelihood of experiencing depression (OR 6.32, 95% CI: 1.61–24.76, p = 0.008). According to the current study, care providers (rather than spouses and family members, OR 1.97, 95% CI: 1.16–3.36, p = 0.012) and living arrangements (not with family members, OR 1.35, 95% CI: 0.45–4.02, p = 0.578) are factors that can predict depression. Individuals who did not receive rehabilitation services had two times greater odds of being depressed (OR 1.85, 95% CI: 1.23–2.77, p = 0.003) than those who received rehabilitation services. Individuals with poor functional status had significantly greater chances of experiencing depression than did those with good functional status (OR 11.03, 95% CI: 7.14–17.04; p <0.001). The information is presented in Table 2.

## Discussion

The findings from this study indicate that 80.8% of participants experienced moderate to severe disability, with 58.1% of participants reporting moderate to extreme levels of depression. Depression was strongly associated with the participants' sex, marital status, monthly family income, length of stroke, medical spending provider status, and care provider status. A previous study conducted in Bangladesh yielded comparable findings, with 70% of individuals exhibiting depressive symptoms and 32% experiencing severe depression [14]. A recent comprehensive analysis indicated that the prevalence of depressive disorders among individuals with stroke varied from 25% to 79%, which aligns with our own findings [23]. In their study, Islam et al. found a substantial correlation between depression and living in a joint family, the inability to independently undertake daily living tasks, and having dysphasia [14]. According to another study conducted in Bangladesh, the prevalence of anxiety and depression was more than twenty times higher among individuals who did not receive rehabilitation services [24]. However, a comprehensive analysis and synthesis of other studies found that the occurrence of depression just after a stroke was 27%, and three months after a stroke it increased to 53%, which is slightly less than what our study observed [25].

PSD was found to have a significant correlation with demographic factors, including being male, married, living in a nuclear family, and residing in an urban area [26]. The present investigation found a substantial correlation between the age, residence, marital status, and educational qualification of the participants and post-stroke depression, which aligns with the findings of previous research. Female gender was identified as a predictor of post-stroke depression by Paolucci et al. in their investigation [27]. Our research confirmed that females had a nearly one-and-a-half-times greater likelihood of developing depression. In contrast to our findings, an Ethiopian study found that males were nearly four times more likely to develop post-stroke depression [17]. The variation in the result may be due to our country's socio-economic condition. Additionally, the study identified a substantial correlation between depression and a diverse array of variables, such as the frequency of strokes, the duration of the stroke, the type of stroke, and the use of nicotine. Comorbidities, medical expenses, and cohabitation situations were also significantly associated with depression. In Northwest Ethiopia, a study demonstrated that poststroke depression was predicted by higher lesion size, stroke severity, history of comorbidity, previous stroke, and physical handicap, which is similar to the findings of the present study [17].

Disability and PSD are interlinked, with a notable correlation between them. This study revealed that stroke patients with low functional abilities are at a twelve-fold increased risk of developing PSD. These findings align with the literature indicating that physical challenges are

**Table 2. Predictors of poststroke depression according to the binary logistic regression model.**

| Predictors | | Odds ratio (95% Cl) | p-value | VIF |
|---|---|---|---|---|
| Gender | Male | Reference | | 2.219 |
| | female | 1.32 (0.71–2.44) | 0.367 | |
| Age group | < 60 years | Reference | | 1.575 |
| | >60 years | 1.17 (0.70–1.4) | 0.532 | |
| Marital status | Unmarried/widow | Reference | | 3.872 |
| | Married | 1.28 (0.42–3.68) | 0.661 | |
| Residence | Urban | Reference | | 1.299 |
| | Rural | 1.28 (0.73–2.25) | 0.375 | |
| Qualification | ≤Primary | Reference | | 1.398 |
| | Secondary or more | 1.08 (0.62–1.88) | 0.782 | |
| Occupation | Unemployed | Reference | | 2.368 |
| | Employed | 1.37(0.76–2.47) | 0.284 | |
| Monthly family income (in taka) | >30000 | Reference | | 1.373 |
| | 15000–30000 | 1.15 (0.70–5.1.91) | 0.566 | |
| | <15000 | 2.39 (1.21–4.71) | **0.011**\* | |
| Tobacco ingestion | No | Reference | | 1.144 |
| | Yes | 1.76 (1.16–2.67) | **0.007**\* | |
| Other concomitant disease | No | Reference | | 1.078 |
| | Yes | 1.68 (1.00–2.82) | **0.047**\* | |
| Type of stroke | Ischemic | Reference | | 1.037 |
| | Hemorrhagic | 1.31 (0.77–2.25) | 0.314 | |
| Onset of stroke | 1st time | Reference | | 1.187 |
| | Recurrent | 3.41 (1.89–6.14) | **<0.001**\*\* | |
| Duration of stroke | < 6 months | Reference | | 1.125 |
| | >6 months | 1.43 (0.84–2.45) | 0.185 | |
| Medical expenses are paid by | Self/spouse | Reference | | 1.395 |
| | Children | -0.68(40–1.13) | 0.140 | |
| | Others | 6.32(1.61–24.76) | **0.00**\* | |
| Care supporter | Spouse and children | Reference | | 1.464 |
| | Others | 1.97(1.16–3.36) | **0.012**\* | |
| Living arrangement | Spouse and children | Reference | | 4.360 |
| | Others | 1.35 (0.45–4.02) | 0.587 | |
| Rehabilitation receiving | Yes | Reference | | 1.115 |
| | No | 1.85(1.23–2.77) | **0.003**\* | |
| MRS category | Good functional status | Reference | | 1.352 |
| | Poor functional status | 11.03 (7.14–17.04) | **<0.001**\*\* | |
| Hosmer and Lemeshow Test | | 0.922 | | |
| Cox & Snell R | | Nagelkerke R | | |
| 36.3–48.9 | | | | |
| Classification Table | | 79.9% | | |

Statistically significant

\***p<0.05**; Statistically significant

\*\***p<001;** VIF–Variance Inflation Factor; CI–Confidence interval

associated with the presence of depressive symptoms in stroke patients [28, 29]. People who are less autonomous in their daily tasks are more prone to exhibit symptoms of PSD following stroke [30]. It is commonly recognized that stroke patients who maintain an active lifestyle are less prone to developing feelings of sadness [28, 31]. However, most stroke patients become inactive and ill and experience a decline in their physical abilities with age [32]. This pattern can lead to a greater likelihood of experiencing recurring depressive signs.

Depressive syndrome was twice as much prevalent among stroke survivors who had not previously received rehabilitation services in this study. Similar to our finding a study conducted in our neighboring country Nepal reported depressive symptoms are three times more prevalent among individuals with disabilities than among the general population [33]. A cross-sectional study in Bangladesh reported that stroke survivors who did not receive rehabilitation treatments had twenty times more anxiety and depression than did those who received these holistic services [24]. Additionally, a Taiwanese population-based survey study indicated that receiving stroke rehabilitation treatment within three months after being admitted for a stroke may considerably lower the chance of developing PSD, consistent with the results of the present study [34]. Rehabilitation services enhance the quality of life of stroke patients, as well as their functional and motor scores. A better emotional state is also a result of improved functional independence due to rehabilitation services [35].

Family function and post-stroke depression are significantly associated. Dysfunctional family acting as a stimulus for triggering post-stroke depression [36]. In this present study, stroke survivors who are living with their spouse and children had significantly lower odds of being developed depression. Wang et al. reported in their study that having a well-functioning family is an important protective factor against post stroke depression [36]. Social support helps to prevent post stroke depression and negatively associated with depression among stroke patients [37]. Relationship with spouse and children improve the overall health and life satisfaction of stroke survivors reported in a nationwide cross-sectional study conducted in China [38]. In this study, stroke survivors who received care and medical expenses from their spouse and children had two- and six-times respectively lower odds of developing depression than those who received care and expenses from others.

A greater degree of depression was observed among the victims who suffered recurrent stroke. A recent systematic review reported that PSD is associated with a significantly increased risk of mortality in stroke survivors [39]. Another meta-analysis suggested that post-stroke depression may be an independent predictor of stroke recurrence among ischemic stroke patients [40]. Participants whose medical expenditure was provided by their children had 35% lower odds of being depressed, and participants whose medical expenditure was provided by their relatives or others had four times greater odds of being depressed. Care providers and cohabitation situations are also predictors of depression, as reported in the present study. Similar to our findings, a cross-sectional survey conducted in the United States reported that annual household income was associated with the risk of poststroke depression [41]. Another systematic review and meta-analysis indicated that social support acted as a protective element against PSD during both the acute and subacute stages [10].

## Importance and clinical significance for public health

This study revealed critical factors that negatively impact PSD among stroke survivors in Bangladesh. The results of this study show how important it is to get help from family members, quit smoking, and prevent recurrent strokes and chronic diseases in order to prevent depression after a stroke. This study's findings will lay the groundwork for what health care experts can do to prevent post-stroke depression. The inclusion of family members and rehabilitation services is essential in preventing post-stroke depression. From a policy standpoint, it is crucial

to involve health professionals, particularly psychiatrists, in order to effectively address post-stroke depression. This should be done by integrating the promotion of family involvement in managing the condition, along with providing rehabilitation services that are equipped with the necessary training and resources. These interventions should be included into national health strategies. **Strengths, limitations and future recommendations**

This study is the first to investigate disability and depression among stroke survivors in Bangladesh. This research aimed to improve data accuracy by using a questionnaire distributed through interviews. Thus, there are a few limitations to this research. This study employed a cross-sectional approach and utilized a nonrandom convenience sampling technique to select participants. The participants comprised only stroke victims who attended three distinct tertiary-level hospitals in Bangladesh. This makes it difficult to generalize the findings to all stroke victims in Bangladesh. We recognize the possible impact of these problems on our study results, but unfortunately, we did not have the resources to fully evaluate their effects. Given these limitations, it is crucial to analyze the results carefully. Future studies on the factors affecting depression in stroke survivors should utilize a longitudinal design and include larger and more diverse sample sizes. This approach facilitated an in-depth exploration of the interconnected attributes associated with disability and depression among stroke victims, thereby broadening and deepening the understanding in this field.

## Conclusion

This study illuminates the key elements of PSD in the specific setting of Bangladesh. The key factors identified included prioritizing family support, financial stability, tobacco cessation, preventing recurrent stroke, and maintaining good functional status and self-dependency. These factors highlight the necessity for specific interventions to address depressive symptoms in stroke patients in this area. The various parameters influencing PSD among stroke victims were clarified in this study. In addition, it offers clinicians, researchers and policymakers' insightful advice on how to advocate for these victims. It is imperative to implement a comprehensive and multidimensional strategy to enhance the competence of life situations for stroke victims, as well as potentially in similar global regions.

## Supporting information

**S1 Checklist. STROBE statement—checklist of items that should be included in reports of observational studies.**
(DOCX)

**S1 Dataset.**
(SAV)

## Acknowledgments

The authors are grateful to all the stroke victims who participated in this study and to the authorities of Sylhet MAG Osmani Medical College Hospital, Parkview Medical College Hospital and Mount Adora Hospital, Sylhet, Bangladesh.

## Author Contributions

**Conceptualization:** Mohammad Jahirul Islam, Sohel Ahmed, Khandaker Md Kamrul Islam, Progya Laboni Tina, Ayon Deb Nath, Nipa Biswas, Md Shafiqul Islam, Bikash Juty Dey Shikder, Muhammed Abdullah Al Mamun, Nasima Yasmin, Shishir Ranjan Chakraborty.

**Data curation:** Mohammad Jahirul Islam, Sohel Ahmed, Bikash Juty Dey Shikder.

**Formal analysis:** Mohammad Jahirul Islam, Sohel Ahmed.

**Investigation:** Sohel Ahmed, Khandaker Md Kamrul Islam, Progya Laboni Tina, Ayon Deb Nath, Nipa Biswas, Md Shafiqul Islam, Bikash Juty Dey Shikder, Muhammed Abdullah Al Mamun, Nasima Yasmin, Shishir Ranjan Chakraborty.

**Methodology:** Mohammad Jahirul Islam, Sohel Ahmed.

**Project administration:** Mohammad Jahirul Islam, Sohel Ahmed, Md Shafiqul Islam.

**Resources:** Bikash Juty Dey Shikder.

**Software:** Mohammad Jahirul Islam, Sohel Ahmed.

**Supervision:** Sohel Ahmed, Shishir Ranjan Chakraborty.

**Validation:** Mohammad Jahirul Islam, Sohel Ahmed, Md Shafiqul Islam, Nasima Yasmin, Shishir Ranjan Chakraborty.

**Visualization:** Sohel Ahmed, Muhammed Abdullah Al Mamun.

**Writing – original draft:** Mohammad Jahirul Islam, Sohel Ahmed.

**Writing – review & editing:** Mohammad Jahirul Islam, Sohel Ahmed, Khandaker Md Kamrul Islam, Progya Laboni Tina, Ayon Deb Nath, Nipa Biswas, Md Shafiqul Islam, Bikash Juty Dey Shikder, Muhammed Abdullah Al Mamun, Nasima Yasmin, Shishir Ranjan Chakraborty.

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
