## [Decision Letter · Decision Letter 0]

18 Jun 2024

PONE-D-24-11475Disability and depression among stroke survivors attending rehabilitation facilities at three designated tertiary care hospitals in Bangladesh: A cross-sectional studyPLOS ONE

Dear Dr. Ahmed,

Thank you for submitting your manuscript to PLOS ONE. After careful consideration, we feel that it has merit but does not fully meet PLOS ONE’s publication criteria as it currently stands. Therefore, we invite you to submit a revised version of the manuscript that addresses the points raised during the review process.

We look forward to receiving your revised manuscript.

Kind regards,

Palash Chandra Banik, MPhil

Academic Editor

PLOS ONE

2. We note that your Data Availability Statement is currently as follows: [All relevant data are within the manuscript and its Supporting Information files]

3. We are unable to open your Supporting Information file [Disability and depression.sav  ]. Please kindly revise as necessary and re-upload.

Reviewers' comments:

Reviewer's Responses to Questions

**Comments to the Author**

1. Is the manuscript technically sound, and do the data support the conclusions?

Reviewer #1: Yes

Reviewer #2: Partly

2. Has the statistical analysis been performed appropriately and rigorously? 

Reviewer #1: Yes

Reviewer #2: No

3. Have the authors made all data underlying the findings in their manuscript fully available?

Reviewer #1: Yes

Reviewer #2: No

4. Is the manuscript presented in an intelligible fashion and written in standard English?

Reviewer #1: Yes

Reviewer #2: No

5. Review Comments to the Author

Reviewer #1: A good study with a sound study design which addresses an important issue in LMICs. Suggest language editing so that it meets international standards. The limitations section can be expanded. The predictors of depression can be analysed more scientifically.

Reviewer #2: Review Comments to the Author

Thank you for the opportunity to review the manuscript titled “Disability and depression among stroke survivors attending rehabilitation facilities at three designated tertiary care hospitals in Bangladesh: A cross-sectional study.” Please see my points below.

Abstract:

1) Revise the background of the abstract as follows” Poststroke depression (PSD) is a highly prevalent and serious mental health condition affecting a significant proportion of stroke survivors worldwide. While its exact causes remain under investigation, managing PSD presents a significant challenge”.

2) Revise “Depression and disability were measured using the Patient Health Questionnaire-9 and Modified Rankin Scale” with “Depression and disability were measured using the Patient Health Questionnaire-9 and the Modified Rankin Scale, respectively”

3) Authors stated “58.1% of them experienced a moderate to significant level of depression”. What does significant means here. It should be severe/extremely severe or something like that.

4) Revise “ Participants whose medical expenses not participants who had.

5) Rewrite the conclusion section. This part should include a brief summary of all the findings along with recommendations based on the data.

Introduction:

1. While the global context is important, consider emphasizing the urgency of understanding depression in Bangladeshi stroke patients due to limited data (lines 74-75).

2. Strengthen the transition by explicitly stating how your study addresses the identified gap (line 76).

3. Consider combining lines 61-63 to improve readability.

Methodology

1. Add reliability statistics for the modified Rankin scale (mRS) and The Patient Health Questionnaire-9 (PHQ-9) for the current study.

2. Add Hosmer–Lemeshow test value for the present study

3. Add VIF values in the respective table as a new column.

4. While the exclusion criteria are clear, some might be subjective (e.g., communication challenges).

5. While experience is mentioned, it would be beneficial to specify if data collectors received training on using the questionnaires and ensuring consistency.

6. The manuscript should address how missing data (if any) was handled during analysis.

7. Further details on the type and duration of rehabilitation services received could be helpful.

Results:

1. In Table 1, note where you performed the Fisher's Exact test and the Chi-square test.

2. The authors reported Prevalence of disability and depression in the "Prevalence of disability and depression among stroke survivors" section. However, I failed to find any supporting tables or figures in the manuscript.

3. Cite the table number in the section on factors associated with depression among stroke survivors.

4. The section titled "Correlation between disability and depression" does not appear to have any supporting tables or figures.

Discussion:

1. The authors compared the prevalence to two studies in the first section of the discussion, but a more comprehensive comparison is required. Also, simply showing similarities with previous studies is insufficient here.

2. The second paragraph of the discussion seems peculiar to me. Authors must identify the significant variables within their study and rigorously compare them to those identified in other relevant studies. This comparative analysis should be closely aligned with the study objectives and framed within the context of existing literature. By delineating the key variables and exploring how they intersect with those identified in prior research, authors can elucidate the underlying mechanisms driving their findings and contextualize their contribution to the field.

3. Lines 256–264 compared the results to numerous other investigations. The authors should provide a thorough explanation for the observed similarities and differences between their findings and those of previous studies.

4. I found some other factors associated with depression and align with study objectives not discussed in the discussion section.

5. In my opinion discussion section need extensive revisions before considering publication in the PLOS ONE.

6. The "Importance and clinical significance for public health" section warrants a more specific and rigorous treatment, augmented by thorough referencing to substantiate its claims. It is essential to provide a comprehensive analysis of the practical implications of the study's findings within the realm of public health. the authors should strive to enrich this section with precise, evidence-based insights that resonate with the overarching objectives of the study and contribute meaningfully to the advancement of public health knowledge and practice.

6. PLOS authors have the option to publish the peer review history of their article (what does this mean?). If published, this will include your full peer review and any attached files.

Reviewer #1: **Yes: **Chathurie Suraweera

Reviewer #2: No

---

## [Author Response · Author response to Decision Letter 0]

18 Jul 2024

Reply to the reviewer’s comments

We express our gratitude to the reviewer for their meticulous and in-depth reading of this manuscript, as well as for their insightful remarks and helpful recommendations, all of which have enabled us to enhance the text's quality.

Reviewer # 1

Comment No: 1 A good study with a sound study design which addresses an important issue in LMICs. Suggest language editing so that it meets international standards. The limitations section can be expanded. The predictors of depression can be analysed more scientifically.

Reply No: 1 We appreciate the insightful feedback and the chance to improve the quality of our manuscript. We have now ensured the correctness of the linguistic issue and expanded the limitation section. The multivariate analysis included the variables that showed significant results in the bivariate study.

Reviewer # 2

We express our gratitude to the reviewer for their meticulous and comprehensive evaluation of this paper, as well as for their insightful remarks and valuable recommendations, which significantly contributed to enhancing the overall quality of this work.

Abstract

Comment No: 1 Revise the background of the abstract as follows” Poststroke depression (PSD) is a highly prevalent and serious mental health condition affecting a significant proportion of stroke survivors worldwide. While its exact causes remain under investigation, managing PSD presents a significant challenge”.

Reply No: 1 Thank you for your insightful comment. We have modified the statement in the revised text.

Comment No: 2 Revise “Depression and disability were measured using the Patient Health Questionnaire-9 and Modified Rankin Scale” with “Depression and disability were measured using the Patient Health Questionnaire-9 and the Modified Rankin Scale, respectively”

Reply No: 2 Thank you for your comment. The revised text has been modified according to your suggestion.

Comment No: 3 Authors stated “58.1% of them experienced a moderate to significant level of depression”. What does significant means here. It should be severe/extremely severe or something like that.

Reply No: 3 Thank you for highlighting our unintentional mistake. We have resolved the issue in the revised text.

Comment No: 4 Revise “Participants whose medical expenses not participants who had.

Reply No: 4 We appreciate your attention in this matter and your contribution to improving the quality of our text. We have corrected the issue in the revised text.

Comment No: 5 Rewrite the conclusion section. This part should include a brief summary of all the findings along with recommendations based on the data.

Reply No: 5 Thank you for your suggestion. As per your recommendation, we have now corrected the conclusion section in the revised text. 

Introduction

Comment No: 1 While the global context is important, consider emphasizing the urgency of understanding depression in Bangladeshi stroke patients due to limited data (lines 74-75).

Reply No: 1 Addressed in the revised text.

Comment No: 2 Strengthen the transition by explicitly stating how your study addresses the identified gap (line 76).

Reply No: 2 Thank you for your comment, The following statement has been added in the revised text: ‘’This study focused on evaluating the prevalence of depression and exploring the relationship between disability and depressive symptoms among Bangladeshi stroke survivors which will help establish the urgency of intervention in this emerging field. Additionally, this study also identified predictors of depression among Bangladeshi stroke survivors. The results are anticipated to play a crucial role in providing valuable insights for both clinical practices and health policy, ultimately enhancing the mental well-being of stroke survivors in Bangladesh.’’

Comment No: 3 Consider combining lines 61-63 to improve readability.

Reply No: 3 Thank you for your suggestion. The following modification has been added in the revised text ‘’The incidence of depression following a stroke range from 25% to 79%, with more than half of these individuals going undiagnosed or untreated.’’

Methodology

Comment No: 1 Add reliability statistics for the modified Rankin scale (mRS) and The Patient Health Questionnaire-9 (PHQ-9) for the current study.

Reply No: 1 In response to your comment, we have added reliability statistics to the revised text for the modified Rankin scale and the Patient Health Questionnaire-9.

Comment No: 2 Add Hosmer–Lemeshow test value for the present study

Reply No: 2 Added in the revised text

Comment No: 3 Add VIF values in the respective table as a new column.

Reply No: 3 Thank you for your comment. We have added a new column to Table 2 that represents the VIF value.

Comment No: 4 While the exclusion criteria are clear, some might be subjective (e.g., communication challenges).

Reply No: 4 Patients with stroke who experienced speech difficulties and were unable to communicate with the data collector during the interview process were excluded.

Comment No: 5 While experience is mentioned, it would be beneficial to specify if data collectors received training on using the questionnaires and ensuring consistency.

Reply No: 5 Thank you for your comment, The following statement is added in the revised text; ‘’Prior to data collection, a training session was conducted to demonstrate data collection process and maintain consistency among the data collectors.’’

Comment No: 6 The manuscript should address how missing data (if any) was handled during analysis.

Reply No: 6 Thank you for highlighting this issue. The following statement has been added in the revised text: ‘’As there was no missing data, this study includes a total of 725 people, including both males and females, participated in this study.’’

Comment No: 7 Further details on the type and duration of rehabilitation services received could be helpful.

Reply No: 7 Participants in this study were those who received physiotherapy and rehabilitation services at three tertiary care hospitals in Sylhet city at least two weeks prior to events. Unfortunately, we did not consider any variables regarding the duration and type of rehabilitation service received.

Results

Comment No: 1 In Table 1, note where you performed the Fisher's Exact test and the Chi-square test.

Reply No: 1 Thank you for your comment. We added the footnote # denoting Fisher’s exact test, and the remaining variables are Chi-square tests.

Comment No: 2 The authors reported Prevalence of disability and depression in the "Prevalence of disability and depression among stroke survivors" section. However, I failed to find any supporting tables or figures in the manuscript.

Reply No: 2 There is no supporting table for the data, but we did include them in the main analysis and represent them in text form.

Comment No: 3 Cite the table number in the section on factors associated with depression among stroke survivors.

Reply No: 3 Thank you for your comment, added in the revised text.

Comment No: 4 The section titled "Correlation between disability and depression" does not appear to have any supporting tables or figures.

Reply No: 4 There is no supporting table for the data, but we did include them in the main analysis and represent them in text form.

Discussion

Comment No: 1 The authors compared the prevalence to two studies in the first section of the discussion, but a more comprehensive comparison is required. Also, simply showing similarities with previous studies is insufficient here.

Reply No: 1 Thank you for your valuable comment which helps to improve the quality of the manuscript. The following lines are added in the revised text: In their study, Islam et al. found a substantial correlation between depression and living in a joint family, the inability to independently undertake daily living tasks, and having dysphasia [14]. According to another study conducted in Bangladesh, the prevalence of anxiety and depression was more than twenty times higher among individuals who did not receive rehabilitation services [24]. However, a comprehensive analysis and synthesis of other studies found that the occurrence of depression just after a stroke was 27%, and three months after a stroke it increased to 53%, which is slightly less than what our study observed [25].

Comment No: 2 The second paragraph of the discussion seems peculiar to me. Authors must identify the significant variables within their study and rigorously compare them to those identified in other relevant studies. This comparative analysis should be closely aligned with the study objectives and framed within the context of existing literature. By delineating the key variables and exploring how they intersect with those identified in prior research, authors can elucidate the underlying mechanisms driving their findings and contextualize their contribution to the field.

Reply No: 2 Thank you for your valuable feedback. We now ensure that the upgraded manuscript fulfills your expectations.

Comment No: 3 Lines 256–264 compared the results to numerous other investigations. The authors should provide a thorough explanation for the observed similarities and differences between their findings and those of previous studies.

Reply No: 3 The revised text has upgraded the statement as per your suggestion.

Comment No: 4 I found some other factors associated with depression and align with study objectives not discussed in the discussion section.

Reply No: 4 Thank you for your in-depth review and for giving us the opportunity to improve the manuscript's quality. As per your recommendation, we now include medical expenses, care provider, and living arrangements in the discussion section. The following statements has been added in the revised text: Family function and post-stroke depression are significantly associated. Dysfunctional family acting as a stimulus for triggering post-stroke depression [37]. In this present study, stroke survivors who are living with their spouse and children had significantly lower odds of being developed depression. Wang et al. reported in their study that having a well-functioning family is an important protective factor against post stroke depression [37]. Social support helps to prevent post stroke depression and negatively associated with depression among stroke patients [38]. Relationship with spouse and children improve the overall health and life satisfaction of stroke survivors reported in a nationwide cross-sectional study conducted in China [39]. In this study, stroke survivors who received care and medical expenses from their spouse and children had two- and six-times respectively lower odds of developing depression than those who received care and expenses from others.

Comment No: 5 In my opinion discussion section need extensive revisions before considering publication in the PLOS ONE.

Reply No: 5 Thank you for your feedback. We now ensure that the upgraded manuscript will satisfy your recommendations.

Comment No: 6 The "Importance and clinical significance for public health" section warrants a more specific and rigorous treatment, augmented by thorough referencing to substantiate its claims. It is essential to provide a comprehensive analysis of the practical implications of the study's findings within the realm of public health. the authors should strive to enrich this section with precise, evidence-based insights that resonate with the overarching objectives of the study and contribute meaningfully to the advancement of public health knowledge and practice.

Reply No: 6 Thank you for your feedback. We now ensure that the upgraded manuscript will satisfy your recommendations.

---

## [Decision Letter · Decision Letter 1]

3 Sep 2024

PONE-D-24-11475R1Disability and depression among stroke survivors attending rehabilitation facilities at three designated tertiary care hospitals in Bangladesh: A cross-sectional studyPLOS ONE

Dear Dr. Ahmed,

Thank you for submitting your manuscript to PLOS ONE. After careful consideration, we feel that it has merit but does not fully meet PLOS ONE’s publication criteria as it currently stands. Therefore, we invite you to submit a revised version of the manuscript that addresses the points raised during the review process.

We look forward to receiving your revised manuscript.

Kind regards,

Palash Chandra Banik, MPhil

Academic Editor

PLOS ONE

Journal Requirements:

Reviewers' comments:

Reviewer's Responses to Questions

**Comments to the Author**

1. If the authors have adequately addressed your comments raised in a previous round of review and you feel that this manuscript is now acceptable for publication, you may indicate that here to bypass the “Comments to the Author” section, enter your conflict of interest statement in the “Confidential to Editor” section, and submit your "Accept" recommendation.

Reviewer #3: All comments have been addressed

2. Is the manuscript technically sound, and do the data support the conclusions?

Reviewer #3: Yes

3. Has the statistical analysis been performed appropriately and rigorously? 

Reviewer #3: Yes

4. Have the authors made all data underlying the findings in their manuscript fully available?

Reviewer #3: Yes

5. Is the manuscript presented in an intelligible fashion and written in standard English?

Reviewer #3: Yes

6. Review Comments to the Author

Reviewer #3: The design, settings, periods and participants are well described

The eligibility criteria are properly addressed

Outcome variables are described and used appropriate tool for data collection. The Statistical methods employed is sufficient. Nevertheless, Authors need to describe how data quality was assured (mention pretesting, standardization of data collection method, was training given etc. The key findings are summarized with reference to study objectives

The strength and limitations of the study is well described, taking into account sources of potential bias or imprecision.

7. PLOS authors have the option to publish the peer review history of their article (what does this mean?). If published, this will include your full peer review and any attached files.

Reviewer #3: **Yes: **Jemal Haidar Ali (Prof)

---

## [Author Response · Author response to Decision Letter 1]

12 Sep 2024

Reply to the reviewer’s comments

We express our gratitude to the reviewer for their meticulous and in-depth reading of this manuscript, as well as for their insightful remarks and helpful recommendations, all of which have enabled us to enhance the text's quality.

Reviewer # 3

We express our gratitude to the reviewer for their meticulous and comprehensive evaluation of this paper, as well as for their insightful remarks and valuable recommendations, which significantly contributed to enhancing the overall quality of this work.

 Title and Abstract

Comment No: 1 Title is clear and addressed an important issue of health significance. Abstract is structured and presented the findings though the world limit exceeded 300.

Reply No: 1 Thank you for your insightful feedback. In compliance with the journal guidelines, we have decreased the word count to 300.

Introduction

Comment No: 2 The scientific background and rationale is well addressed. Objectives Clear and concise

Reply No: 2 Thank you for your valuable comment

Methodology

Comment No: 3

The design, settings, periods and participants are well described

The eligibility criteria are properly addressed

Outcome variables are described and used appropriate tool for data collection. The Statistical methods employed is sufficient

Nevertheless, Authors need to describe how data quality was assured (mention pretesting, standardization of data collection method, was training given etc…

Reply No: 3 Thank you for your positive feedback. The following statements has been added in the revised text: The interviews were performed by three proficient data collectors, all of whom were physiotherapy graduates with more than three years of experience. Prior to data collection, a training session was conducted to demonstrate data collection process and maintain consistency among the data collectors. In order to address any issues that may arise during data collection, each data collector carried out a pre-test session including a minimum of five data. These data were excluded from the final analysis.

Results

Comment No: 4 Surprising to see 100.0% response rate? How this was possible given the study is facility based and sometimes we see missing information? Elaborate on this further to ensure the credibility of the work.

Report numbers of individuals including the percentages and use appropriate scientific language

Reply No: 4 We appreciate you posing this inquiry and affording us the chance to provide reasonable justifications. Our data collection process utilized the echo-friendly Google Forms platform as an alternative to traditional pen and paper methods. The inclusion of the mandatory option on each question serves to avoid incomplete submissions.

Discussion

Comment No: 5 The key findings are summarized with reference to study objectives

The strength and limitations of the study is well described, taking into account sources of potential bias or imprecision.

Reply No: 5 Thank you for your positive feedback.

Other information

Comment No: 6 Authors need to acknowledge all who assisted/participated in tins study

Reply No: 6 That material has already been provided in the acknowledgement section.

Comment No: 7 Mention the contribution of authors

Reply No: 7 Your insightful remark is appreciated. The information has already been incorporated into the journal system in accordance with the prescribed guidelines of the journal.

Comment No: 8 Interestingly, authors have attached the raw data

Reply No: 8 Raw data was included as per journals requirement

---

## [Editor Report · Decision Letter 2]

17 Sep 2024

Disability and depression among stroke survivors attending rehabilitation facilities at three designated tertiary care hospitals in Bangladesh: A cross-sectional study

PONE-D-24-11475R2

Dear Dr. Ahmed,

We’re pleased to inform you that your manuscript has been judged scientifically suitable for publication and will be formally accepted for publication once it meets all outstanding technical requirements.

Kind regards,

Palash Chandra Banik, MPhil

Academic Editor

PLOS ONE
---

## [Editor Report · Acceptance letter]

23 Dec 2024

PONE-D-24-11475R2 

PLOS ONE

Dear Dr. Ahmed, 

I'm pleased to inform you that your manuscript has been deemed suitable for publication in PLOS ONE. Congratulations! Your manuscript is now being handed over to our production team.

Kind regards, 

on behalf of

Dr. Palash Chandra Banik 

Academic Editor

PLOS ONE